# Bioactive Compounds, Amino Acids, Fatty Acids, and Prebiotics in the Seed of Mahuad (*Lepisanthes rubiginosa* (Roxb.) Leenh)

**Apichaya Bunyatratchata [1,2], Theeraphan Chumroenphat [2,3], Surapon Saensouk [4]**
**and Sirithon Siriamornpun [1,2,*]**

1 Department of Food Technology and Nutrition, Faculty of Technology, Mahasarakham University, Kantarawichai, Maha Sarakham 44150, Thailand; apichaya.b@msu.ac.th
2 Research Unit of Thai Food Innovation (TFI), Mahasarakham University, Kantarawichai, Maha Sarakham 44150, Thailand; theeraphan.c@ubru.ac.th
3 Aesthetic Sciences and Health Program, Faculty of Thai Traditional and Alternative Medicine, Ubon Ratchathani Rajabhat University, Mueang District, Ubon Ratchathani 34000, Thailand
4 Walai Rukhavej Botanical Research Institute, Mahasarakham University, Kantarawichai, Maha Sarakham 44150, Thailand; surapon.s@msu.ac.th
* Correspondence: sirithon.s@msu.ac.th; Tel.: +66-43-754085

**Abstract:** The seeds of Mahuad (*Lepisanthes rubiginosa* (Roxb.) Leenh (LRL) were analyzed for proximate composition and the contents of phenolic acids, flavonoids, and sugars/oligosaccharides. The LRL seeds contained approximately 29% moisture, 10% protein, 2% fat, 16% fiber, 2% ash, and 42% carbohydrate. The major phenolic acids were vanillic acid and *p*-hydroxybenzoic acid, accounting for 30% and 26% of total phenolic content, respectively. The predominant flavonoids were quercetin (62% of total flavonoid content) followed by myricetin (22%). Proline, methionine, and arginine were the dominant amino acids, constituting 35%, 19%, and 13% of total amino acid content, respectively. Prebiotic fructooligosaccharide (5.3 mg/g) and stachyose (4.2 mg/g) were also found in the LRL seeds. The major fatty acids were palmitic acid (C 16:0, 41%), oleic acid (C 18:1n9, 27%), and linoleic acid (C 18:2n6, 19%). This information reveals useful information about LRL seeds as a potential source of bioactive compounds for future use in various aspects including food, feeds, pharmaceuticals, and cosmetics.

**Keywords:** wild seed; vanillic acid; *p*-hydroxybenzoic acid; quercetin; fructooligosaccharide




## 1. Introduction

Mahuad (*Lepisanthes rubiginosa* (Roxb.) Leenh) (LRL) is a Southeast Asian plant that is widely grown in Thailand and in other Southeast Asian countries such as Indonesia, India, and Malaysia [1]. This plant has a long history of use in folk medicine to prevent itching, suppress or relieve coughing, and treat headaches, fevers, etc. [1,2]. Several bioactive compounds have been isolated and reported from each part of LRL. For instance, alkaloids, flavonoids, phenolics, tannins, and saponin were found in leaves, and bark also contained alkaloids, saponins, and phenolic compounds [2,3]. The oil extracted from flowers and fruits also contained several organic acids [4]. Numerous beneficial effects such as antioxidant, anticancer, analgesic, antibacterial, antihyperglycemic, neuropharmacological, and antidiarrheal activities have been demonstrated in LRL [2–4]. The phytochemicals found in this plant may be associated with these health benefits.

LRL fruit, which is seasonal, is generally consumed in its ripe stage. In Thailand, this fruit has long been widely consumed by local Thai people, sometimes being consumed along with its seed. Our previous study focused on the bioactive compounds present in LRL fruit pulp, demonstrating its potential as a source of bioactive compounds, particularly lycopene, cyanidin-3-O-glucoside, and quinic acid [5]. However, the bioactive compounds in LRL seeds have not been thoroughly explored. It is worth noting that the fruits belonging

to the genus *Lepisanthes* contain one to four ellipsoid seeds [6], with the seed of LRL fruit representing a significant portion of the fruit. Although there has been limited prior research on LRL seeds, a recent study on seed extracts from a related species, *Lepisanthes alata* Leenh, demonstrated high levels of total phenolic content, total flavonoid content, and antioxidant activity, more than the peel and the flesh extracts [7]. This discovery has raised further interest in exploring LRL seeds and their potential applications, as they may contain valuable phytochemicals similarly to other parts of LRL that were previously reported.

Hence, the aim of the present study was to extend knowledge by investigating the bioactive compounds present in LRL seeds, including phenolic acids, flavonoids, amino acids, sugars/oligosaccharides, and fatty acids. The investigation also included a proximate analysis of LRL seeds for a comprehensive understanding of their composition. This plant is included in the Plant Genetic Conservation Project Under the Royal Initiative of Her Royal Highness Princess Maha Chakri Sirindhorn (RSPG). Consequently, there is a possibility the fruit and seeds will be used more extensively, potentially leading to various projects aimed at promoting the cultivation of these plants and the development of LRL-based products to support local communities in the future. Importantly, prior to our research, there had been no studies focusing on quantifying individual compounds and establishing comprehensive profiles of LRL seeds. To our knowledge, this research is the first to report on individual composition of phytochemicals in LRL seeds. We expect to generate valuable information about this fruit seed to explore its potential in various fields and applications.

## 2. Materials and Methods

### 2.1. Chemicals and Reagents

In this research, the solvents and reagents used were analytical grade and the solvents employed for chromatography analysis were HPLC grade. The standards were purchased from Sigma-Aldrich Co. (St. Louis, MO, USA). The solvents and reagents used in both HPLC analysis and antioxidant analysis were obtained from Merck (Darmstadt, Germany). The set of phenolic acid standards consisted of gallic acid, protocatechuic acid, p-hydroxybenzoic acid, vanillic acid, syringic acid, vanillin, p-coumaric acid, ferulic acid, sinapic acid, cinnamic acid, and gentisic acid. Additionally, we employed four flavonoid standards including rutin, quercetin, apigenin, and myricetin. For determination of amino acids, a total of 20 standards were utilized, including arginine, histidine, isoleucine, leucine, lysine, methionine, phenylalanine, threonine, tryptophan, valine, alanine, asparagine, aspartic acid, cysteine, glutamine, glutamic acid, glycine, proline, serine, and tyrosine. The sugars/oligosaccharides under investigation included stachyose, fructooligosaccharide, sucrose, glucose, fructose, mannitol, and sorbitol. The chemicals used in this study were analytical grade and included 2,2-diphenyl-1-picrylhydrazyl (DPPH), Folin–Ciocalteu's reagent, sodium carbonate ($NaCO_3$), sodium nitrite ($NaNO_2$), aluminum chloride hexahydrate ($AlCl_3 \cdot 6H_2O$), hexane, acetone, ethanol, and hydrochloric acid.

### 2.2. Sample Preparation

LRL seeds were obtained from LRL fruits collected from the northeastern region of Thailand during April and May 2021. The botanical identification of these plants was conducted by plant taxonomists from Walai Rukhavej Botanical Research Institute at Mahasarakham University, Thailand. The specimens were deposited in the herbarium, and the voucher specimen number was CT020422. After washing the fruits, the seeds were manually removed from the fruit pulp. The seeds were carefully washed with distilled water and subsequently freeze-dried using a Scanvac CoolSafe model 100-9 Pro freeze-dryer (LaboGene ApS, Lillerød, Danmark). The drying process continued until the moisture content of the seeds reached below 7%. The samples were ground and passed through a 40-mesh wire sieve. The freeze-dried samples were stored at −20 °C until they were ready for analysis of phenolic acids, flavonoids, amino acids, sugars/oligosaccharides, and fatty acids.



### 2.3. Proximate Analysis

Fresh seeds after washing were ground and passed through a 40-mesh wire sieve. Subsequently, an analysis of the seed powder was conducted to measure moisture, protein (Kjeldahl method), fat (Soxhlet extraction), fiber, and ash content. The quantification of these parameters was performed using methods outlined in AOAC (2000) [8]. The content of carbohydrate was calculated by difference method using the following equation:

$$\text{Carbohydrate (\%)} = 100 - (\%\text{Moisture} + \%\text{Protein} + \%\text{Fat} + \%\text{ Ash} + \%\text{Fiber}).$$

The experiment was performed in triplicate, and the results are expressed as the mean ± one standard deviation (SD).

### 2.4. Quantification of Individual Phenolic Acids and Flavonoids by HPLC

The extraction of phenolic acids and flavonoid compounds was performed following the procedure detailed in previously published work [5]. Samples (1.0 g) were extracted for 12 h at 37 °C. This extraction was carried out using 20 mL of HCl/methanol (1:100, *v/v*) with continuous shaking at 150 rpm in the dark. After the extraction process, the solution was filtered and the pellet was re-extracted. The filtrates from both extractions were combined and dried under vacuum at 40 °C using a rotary evaporator (Büchi R-210; Flawil, Switzerland). The residue was re-dissolved in 5 mL of methanol/water (50:50, *v/v*). The extracted samples were subjected to high performance liquid chromatography (HPLC) analysis (Shimadzu, Kyoto, Japan). The separation of compounds was achieved using a C18 column (Inertsil® ODS-3; 250 mm × 4.6 mm i.d., 5 μm, GL Sciences Inc., Tokyo, Japan) following the protocol as described in a previous publication [9]. The extracted samples were identified using external standards for phenolic acids and flavonoids. The samples were analyzed in three replicates, and the results are expressed as μg/g dry basis (db) of the sample.

### 2.5. Quantification of Amino Acids by LC/MS/MS

Amino acid extraction was conducted following a previously described procedure [10]. Briefly, 100 mg of the sample were combined with 1 mL of a 0.5 M HCl–ethanol solution (ratio of 1:1 *v/v*). The mixture was vortexed for 2 min, followed by centrifugation at 12,000× *g* for 15 min at 4 °C. The supernatant was collected and filtered through a 0.22 μm nylon membrane. The filtered sample was subjected to analysis using liquid chromatography–mass spectrometry (LC/MS/MS). This analysis was carried out using a Shimadzu LCMS-8030 triple-quadrupole mass spectrometer (Kyoto, Japan) coupled with ESI mode and a Shimadzu LC-20AC series HPLC system. The analysis of amino acids was conducted using isocratic elution on an InertSustain® C18 column (2.1 × 150 mm, 3 μm), along with a guard column. The mobile phase consisted of solvent A (0.1% *v/v* formic acid in water) and solvent B (0.1% *v/v* formic acid in a water/methanol mixture; 50:50 *v/v*). The flow rate was maintained at 0.2 mL/min and the column temperature was at 40 °C. For MS/MS, parameters were set with a nebulizing gas flow rate of 3 L/min at the interface, a DL temperature of 250 °C, a heat block temperature of 400 °C, and a 15 L/min drying gas flow rate [10]. The extracted samples underwent identification using external standards for amino acids. The samples were analyzed in three replicates, and the results were quantified in terms of amino acid content, expressed as μg/g dry basis (db) of the sample.

### 2.6. Quantification of Sugars and Oligosaccharides by HPLC

The extraction of sugars and oligosaccharides was performed as follows: 0.1 g of powdered dry samples was mixed with 5 mL of distilled water. Subsequently, this mixture was incubated at 80 °C for 30 min. After incubation, the samples underwent filtration through a 0.22 mm filter. The filtrates were then subjected to analysis for individual monosaccharides (glucose, fructose, mannitol, sorbitol), sucrose, and oligosaccharides (stachyose, fructooligosaccharide) using the Shimadzu HPLC 20 series system (Shimadzu, Kyoto,

Japan) following a previously established procedure [11]. The quantification of sugars and oligosaccharides were conducted using an Aminex HPX 87C column (300 mm × 7.8 mm; particle size 9 µm, Bio-Rad, Marnes-la-Coquette, France) coupled with a refractive index detector. The samples were analyzed in three replicates, and the results are expressed as mg/g dry basis (db) of the sample.

### 2.7. Quantification of Fatty Acids

The samples were submitted to the Central Laboratory (Khon Kaen, Thailand) for fatty acid composition and concentration analysis. The samples were analyzed according to Association of Official Analytical Chemists (AOAC) method 996.06 [12].

### 2.8. FTIR Measurements

The spectrum of LRL seed samples was obtained using Fourier transform infrared (FTIR) spectroscopy. This analysis was performed according to a previously published method [13] with a UATR accessory for Frontier equipped with a Diamond/KRS-5crystal composite (Perkin Elmer, Waltham, MA, USA).

## 3. Results

### 3.1. Proximate Analysis

The contents of moisture, protein, fat, fiber, ash, and carbohydrate in the LRL seeds are shown in Table 1. Carbohydrates were identified as the predominant constituent, comprising 41.6% of the seed's composition. Following carbohydrates, the moisture content represented 29.1%, fiber 15.9%, protein 9.6%, fat 2.1%, and ash 1.7% of the seed's composition, respectively.

**Table 1.** Proximate composition of *Lepisanthes rubiginosa* (Roxb.) Leenh seed.

| Parameter | *Lepisanthes rubiginosa* (Roxb.) Leenh Seed |
| :---: | :---: |
| Moisture (%) | 29.1 ± 0.5 |
| Crude Protein (%) | 9.6 ± 0.2 |
| Crude Fat (%) | 2.1 ± 0.0 |
| Crude Fiber (%) | 15.9 ± 0.2 |
| Ash (%) | 1.7 ± 0.1 |
| Carbohydrate (%) | 41.6 ± 0.3 |

Values are expressed as means ± SD of triplicate measurements (*n* = 3).

### 3.2. Quantification of Individual Phenolic Acids and Flavonoids in LRL Seed

Table 2 presents the comprehensive analysis of individual phenolic acids and flavonoids in LRL seeds. Among 13 monitored phenolic acids, vanillic acid and *p*-hydroxybenzoic acid emerged as the most abundant compounds in the LRL seeds, accounting for approximately 30% and 26% of total phenolic acid content, respectively (Table 2). The predominant flavonoids observed were quercetin followed by myricetin. Specifically, quercetin comprised approximately 62% of the total flavonoid content, while myricetin accounted for approximately 22%.

**Table 2.** Contents of phenolic acids and flavonoids in *Lepisanthes rubiginosa* (Roxb.) Leenh seed.

| Parameter | Concentration (µg/g) | Percentage (%) |
| :---: | :---: | :---: |
| *Phenolic acids* | | |
| gallic acid | 6.6 ± 0.1 | 11.9 |
| protocatechuic acid | 0.2 ± 0.0 | 0.4 |
| *p*-hydroxybenzoic acid | 14.3 ± 0.2 | 25.7 |
| vanillic acid | 16.9 ± 0.5 | 30.4 |
| syringic acid | 5.0 ± 0.1 | 9.0 |
| vanillin | 6.8 ± 0.2 | 12.2 |
| *p*-coumaric acid | 0.6 ± 0.0 | 1.1 |

**Table 2.** *Cont.*

| Parameter | Concentration (µg/g) | Percentage (%) |
|---|---|---|
| ferulic acid | 2.6 ± 0.1 | 4.7 |
| sinapic acid | 1.1 ± 0.1 | 2.0 |
| cinnamic acid | 0.8 ± 0.0 | 1.4 |
| gentisic acid | 0.7 ± 0.0 | 1.3 |
| *Total phenolic acids* | 55.6 ± 0.9 | 100 |
| *Flavonoids* | | |
| rutin | 3.5 ± 0.1 | 3.8 |
| quercetin | 57.5 ± 0.3 | 62.0 |
| apigenin | 11.1 ± 0.1 | 12.0 |
| myricetin | 20.6 ± 0.0 | 22.2 |
| *Total flavonoids* | 92.7 ± 0.5 | 100 |

Values are expressed as mean ± SD of triplicate measurements (*n* = 3).

### 3.3. Profile and Content of Amino Acids in LRL Seeds

The amino acid composition of the LRL seeds, encompassing both essential and non-essential amino acids, is comprehensively presented in Table 3. Among the amino acids investigated, proline exhibited the highest concentration with a level of 437.5 µg/g. Following proline, methionine and arginine were observed at levels of 230.7 µg/g and 164.3 µg/g, respectively. These three amino acids contributed 35%, 19%, and 13% of total amino acid content in the LRL seeds, respectively. The chromatogram of amino acids in both the standard and LRL seeds is presented in the supplementary materials (Figure S1 in Supplementary Materials). These findings provide valuable insights into the amino acid profile of LRL seeds and emphasize the importance of these amino acids in the nutritional composition.

**Table 3.** Contents of amino acids in *Lepisanthes rubiginosa* (Roxb.) Leenh seeds.

| Amino Acids | Concentration (µg/g) | Percentage (%) |
|---|---|---|
| *Essential AA* | | |
| arginine | 164.3 ± 11.4 | 13.3 |
| histidine | 5.0 ± 0.2 | 0.4 |
| isoleucine | 7.3 ± 0.4 | 0.6 |
| leucine | 15.2 ± 0.7 | 1.2 |
| lysine | 24.6 ± 1.1 | 2.0 |
| methionine | 230.7 ± 6.6 | 18.6 |
| phenylalanine | 7.1 ± 0.3 | 0.6 |
| threonine | 9.8 ± 0.7 | 0.8 |
| tryptophan | 8.1 ± 0.3 | 0.7 |
| valine | 43.7 ± 1.0 | 3.5 |
| *Total essential AA* | 515.8 ± 22.6 | 41.6 |
| *Nonessential AA* | | |
| alanine | 37.9 ± 2.3 | 3.1 |
| asparagine | 4.1 ± 0.2 | 0.3 |
| aspartic acid | 69.2 ± 2.4 | 5.6 |
| cysteine | ND | ND |
| glutamine | 19.2 ± 1.0 | 1.6 |
| glutamic acid | 72.7 ± 0.9 | 5.9 |
| glycine | 0.3 ± 0.0 | 0.0 |
| proline | 437.5 ± 5.6 | 35.3 |
| serine | 9.5 ± 0.5 | 0.8 |
| tyrosine | 72.4 ± 0.8 | 5.8 |
| *Total nonessential AA* | 722.8 ± 13.8 | 58.4 |
| ***Total amino acids*** | 1238.6 ± 14.5 | 100 |

Values are expressed as mean ± SD of triplicate measurements (*n* = 3); ND: Not detected.

### 3.4. Sugar and Oligosaccharide Content in LRL Seeds

Table 4 presents the quantification of individual sugars and oligosaccharides, namely fructooligosaccharides (FOS) and stachyose, in the LRL seeds. FOS, which are known prebiotics, naturally occur in various plant sources such as onion, artichoke, and chicory [14]. Similarly, stachyose, also recognized as a prebiotic, has been identified in plants and has demonstrated the potential in modulating gut microbiota [15,16]. A prebiotic is defined as "a substrate that is selectively utilized by host microorganisms conferring a health benefit" [17]. The analysis revealed that LRL seeds contain approximately 5.3 mg/g of FOS and 4.2 mg/g of stachyose, highlighting their potential as valuable prebiotic sources. The HPLC chromatogram of sugars and oligosaccharides in LRL seeds is presented in the Supplementary Materials (Figure S2).

**Table 4.** Contents of sugars and oligosaccharides in *Lepisanthes rubiginosa* (Roxb.) Leenh seeds.

| Individual Sugars | Contents (mg/g) |
|---|---|
| Stachyose | $4.2 \pm 0.1$ |
| Fructooligosaccharide | $5.3 \pm 0.0$ |
| Sucrose | $93.5 \pm 0.2$ |
| Glucose | $65.0 \pm 0.1$ |
| Fructose | ND |
| Mannitol | ND |
| Sorbitol | ND |

Values are expressed as mean $\pm$ SD of triplicate measurements ($n$ = 3); ND: Not detected.

### 3.5. Profile and Content of Fatty Acids in LRL Seeds

The fatty acid composition and concentrations identified by gas chromatography (GC) are summarized in Table 5. The fatty acid profiles of LRL were characterized with respect to saturated fatty acids (SFA), monounsaturated fatty acids (MUFA), and polyunsaturated fatty acids (PUFA). The results revealed that the predominant SFA was palmitic acid (C 16:0, 41%), followed by MUFA, oleic acid (C 18:1n9, 27%), and PUFA, linoleic acid (C 18:2n6, 19%). The LRL seed exhibited a fatty acid ratio of 2.3:1.6:1.0 for SFA:MUFA:PUFA, respectively. It has been suggested that the composition of fatty acids in food and the relative balance of different fatty acids are considered to be of greater significance than their quantity [18].

**Table 5.** Contents of fatty acids in *Lepisanthes rubiginosa* (Roxb.) Leenh seeds.

| Fatty Acids | Concentration (g/100g) | % Composition |
|---|---|---|
| *Saturated fatty acids* | | |
| Palmitic acid (C16:0) | 0.30 | 41.10 |
| Stearic acid (C18:0) | 0.03 | 4.11 |
| Behenic acid (C22:0) | 0.01 | 1.37 |
| *Total saturated fatty acids (SFA)* | 0.34 | 46.58 |
| *Unsaturated fatty acids* | | |
| cis-9-Oleic acid (C18:1n9c) | 0.20 | 27.40 |
| cis-11-Eicosenoic acid (C20:1n11) | 0.04 | 5.48 |
| *Total monounsaturated fatty acids (MUFA)* | 0.24 | 32.88 |
| cis-9,12-Linoleic acid (C18:2n6) | 0.14 | 19.18 |
| cis-11,14-Eicosadienoic acid (C20:2) | 0.01 | 1.37 |
| *Total polyunsaturated fatty acid (PUFA)* | 0.15 | 20.55 |
| *Total unsaturated fatty acids* | 0.39 | 53.42 |
| **Total fatty acids** | **0.73** | **100** |

### 3.6. Spectrum of LRL Seed by FTIR

FTIR spectroscopy analysis is a valuable analytical technique widely used to identify functional groups and linkage bonds in polysaccharides [19]. In this study, the FTIR spectrum of LRL seeds was obtained within the range of 400–4000 cm$^{-1}$ (Figure 1). The

spectral analysis revealed a significant peak between 3375 and 3320 cm$^{-1}$. Within the 3400–3200 cm$^{-1}$ range, there were symmetric and asymmetric stretching vibrations of polymetric hydroxyl groups (O–H) and H-bonded stretching, which are characteristic of polyphenolic compounds. The spectral region ranging from 2940 to 2925 cm$^{-1}$ revealed stretching vibrations corresponding to –CH, –CH$_2$, and –CH$_3$, originating from carbohydrates and sugars. The C–H and C=C–C ring-related vibrations suggested the presence of one or more aromatic rings in the chemical compound's structure [20]. Stretching vibrations of the C–H and C=C–C aromatic bonds are observed in the spectral range at 1640 cm$^{-1}$, aiding in flavonoid identification [21]. Additionally, the stretching related to the phenolic C–O bond was detected at approximately 1200 cm$^{-1}$, associated with the C–O configuration found in a pyran ring, a typical feature of the flavonoid C-ring [20]. Furthermore, in the 1200–950 cm$^{-1}$ region, several intense peaks were observed, possibly corresponding to vibrations of glycosidic bonds and pyranoid rings [22,23]. These characteristic peaks in the FTIR spectrum create a unique "fingerprint" for LRL seed.

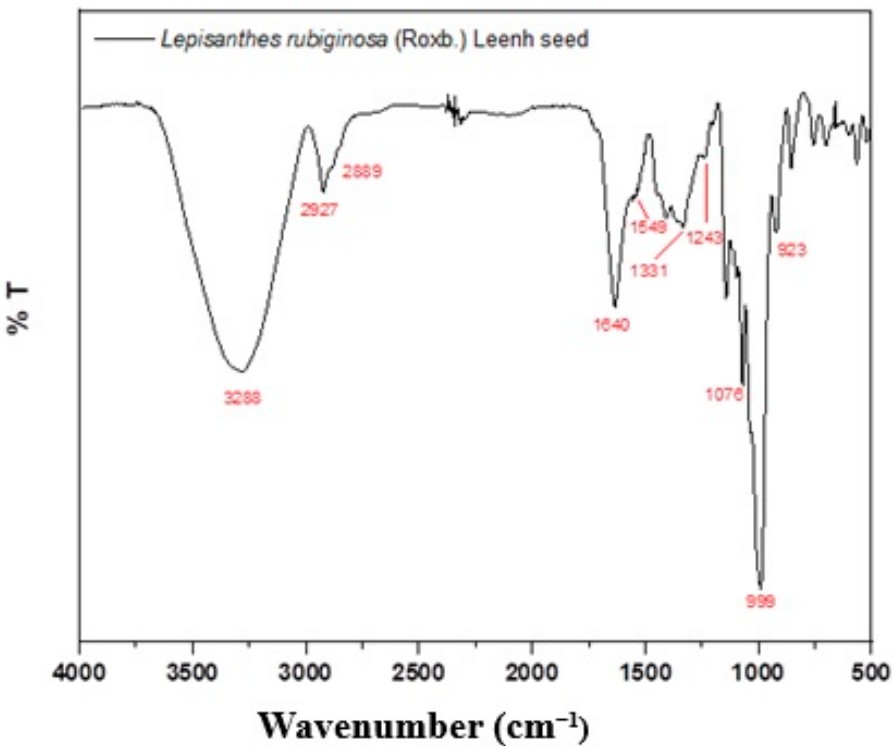

**Figure 1.** The FTIR spectrum of *Lepisanthes rubiginosa* (Roxb.) Leenh seed.

## 4. Discussion

The compositional analysis presented in Table 1 reveals that the LRL seeds exhibit a higher fiber content compared to several other seeds such as watermelon seeds (ranging from 4–7% depending on the varieties) [24], *Grewia tenax* or Guddaim (13%) [25], and mango seed kernel (3-4% depending on varieties) [26]. Furthermore, when compared to major cereals including wheat, rye, corn, barley, oat, rice, millet, and sorghum [27], it was observed that LRL seeds exhibited higher fiber levels but lower carbohydrate content. Typically, LRL seeds have a lower carbohydrate content compared to these cereals, which generally fall within the range of 56.2–81.5% [28,29]. Conversely, LRL seeds demonstrated a higher fiber content compared to these cereals, which typically range from 2.24% to 13.2% [28–30]. This high fiber content observed in LRL seeds might potentially serve as feed or food additives for both humans and animals. In terms of protein and fat content, LRL seeds exhibited similar ranges as many cereals. The protein content of the LRL seed was comparable to rice (7.7%), corn (8.8%), and rye (9.4%) [29]. Similarly, the fat contents of the LRL seed fell within ranges similar to wheat (1.8%), rye (1.7%), barley (2.1%), and

rice (2.2%) [29]. These findings contribute valuable insights into the nutritional aspects of utilizing LRL seeds, highlighting their potential as a viable option in enhancing dietary fiber intake. Further investigation and studies are needed to explore the full nutritional potential, possible health benefits and toxicity associated with the incorporation of LRL seeds into human or animal diets.

The investigation of bioactive compounds in LRL seeds focused on individual phenolic acids and flavonoids (Table 2). Among the identified compounds, vanillic acid emerged as a significant phenolic compound in the LRL seeds, which is commonly found in various food sources and plants [31]. This particular compound serves not only as a flavoring agent but also exhibits many health effects, including antioxidant, anti-inflammatory, anticancer, antidiabetic, antiobesity, and antibacterial activities [31,32]. Our analysis revealed a vanillic acid content of 16.9 µg/g in LRL seeds, which closely aligns with the levels observed in fresh date varieties (14.5–21.3 µg/g) [33]. Additionally, the second most abundant phenolic acid in LRL seeds was *p*-hydroxybenzoic acid, known for its application as a preservative in various cosmetic and pharmaceutical products such as shampoos, moisturizers, and toothpaste [34].

The predominant flavonoids identified in LRL seeds were quercetin and myricetin (Table 2). Quercetin, a powerful antioxidant stronger than vitamins C and E, is commonly found in various foods such as fruits, vegetables, wine, and tea [35,36]. This flavonoid compound is associated with diverse health benefits, including the potential prevention of diseases such as osteoporosis, cancer, and cardiovascular diseases [35]. The content of quercetin in LRL seed was determined to be 57.5 µg/g, surpassing the levels observed in apples (40.1 µg/g) [37]. Similarly, myricetin, another common flavonoid found in plants, exhibits several health benefits, including antioxidant, anticancer, anti-inflammation, and antidiabetic activities [38]. Emerging scientific evidence also suggests potential associations between myricetin and disease prevention, particularly for Parkinson's and Alzheimer's diseases [38]. Myricetin can also act as a preservative to protect against lipid oxidation [38]. Given the presence of these beneficial phenolic acids and flavonoids in LRL seeds, there is potential for their application as functional ingredients in various aspects including food, pharmaceuticals, and cosmetics. The significance of these findings may pave the way for novel avenues in product development, offering opportunities for enhancing human health and well-being.

The comprehensive analysis of amino acids in LRL seeds revealed the prevalence of proline, methionine, and arginine (Table 3). Proline is involved in numerous biological processes, including protein and amino acid synthesis, structure, nutrition, and immune response [39]. Methionine, on the other hand, plays important roles in metabolism, the immune system, antioxidant, and digestive function [40,41]. Cereals and legumes serve as primary food and feed sources for both humans and animals; however, they often lack adequate levels of certain essential amino acids, especially methionine and lysine [42]. In poultry diets, methionine stands out as an essential amino acid and is typically the first limiting amino acid [43–45]. This sulfur-containing amino acid is also a limiting amino acid in other animal feeds including fish and deer [41,46]. Arginine participates in diverse biological activities, such as protein synthesis, immune response, and nitric oxide production [47]. Both proline and arginine are major amino acids in animal-derived feeds [48]. The exploration of amino acid profiles in LRL seeds offers potential applications in nutraceuticals for both animal and human nutrition.

Prebiotic FOS or oligofructose has been the subject of numerous studies, revealing its potential benefits in enhancing mineral absorption, reducing cholesterol levels, and demonstrating a prebiotic effect [14]. This study assessed the FOS content in LRL seeds, revealing a concentration of approximately 5.3 mg/g db (Table 4). This value surpasses the FOS/oligofructose levels found in numerous food or feed ingredients including barley (averaging 1.92 mg/g db), oat (0.36 mg/g db), peanut hulls (2.40 mg/g db), rice bran (0.14 mg/g db), soybean hulls (0.12 mg/g db), alfalfa meal (2.24 mg/g db), wheat (1.36 mg/g db), wheat germ (4.68 mg/g db), and wheat bran (4.00 mg/g db) [49]. In

addition to prebiotic FOS, stachyose is also present in LRL seeds, with a concentration of approximately 4.2 mg/g db (Table 4). In vivo studies have demonstrated that stachyose could promote the growth of beneficial microbiota and increase bacterial diversity [16]. Moreover, animal studies on prebiotic consumption have revealed improvements in the gut microbial community and stool quality [49]. Given these findings, LRL seeds emerge as a potential source of prebiotics that can modulate gut microbiota, offering health benefits to both humans and animals.

The examination of the fatty acid composition of LRL seeds is crucial for evaluating their nutritional value. One important aspect to note is that the fatty acid composition of food holds greater significance than its content [18]. The optimum ratio of saturated fatty acids (SFA) to monounsaturated fatty acids (MUFA) to polyunsaturated fatty acids (PUFA) has been reported to be 1:1:1 [50]. In the case of LRL seeds, the ratio of saturated, monounsaturated, and polyunsaturated fatty acids was found to be 2.3:1.6:1 (Table 5). It is important to emphasize that the optimal ratios of SFA:MUFA:PUFA vary among humans, animals (monogastric or ruminants), and plants [51]. The PUFA:SFA ratio typically $\geq 0.4$ is often considered a "health index" for fatty acid-rich foods or diets [52]. In this analysis of LRL seeds, the PUFA/SFA ratio is approximately 0.44, which falls in line with the recommended value (Table 5). Two essential fatty acids for humans are linoleic acid and alpha-linolenic acid. In LRL seeds, linoleic acid makes up approximately 19% of the total fatty acid content, which is similar to the proportion found in filbert nut (17%) [53].

FTIR spectroscopy is a valuable analytical tool for identifying functional groups and linkage bonds within polysaccharides [19]. In this study, the FTIR spectrum of LRL seeds revealed various functional groups associated with polyphenolic compounds, carbohydrates, sugars, and flavonoids as well as glycosidic bonds and pyranoid rings (Figure 1). This distinctive FTIR spectrum serves as a "fingerprint" of the LRL seed, facilitating its discrimination from other types of seeds or materials. The functional groups identified through FTIR analysis also serve as additional confirmation for the compounds identified in the HPLC results. The presence of these specific functional groups and/or linkages provides valuable insights into the chemical composition of LRL seeds, contributing to a deeper understanding of their properties.

## 5. Conclusions

According to our analytical data, this study is the first to provide a detailed analysis of the individual composition of bioactive compounds in LRL seeds. The analysis of LRL seeds revealed their potential by highlighting their abundance in fiber and essential compounds including vanillic acid and *p*-hydroxybenzoic acid as major phenolic acids, quercetin and myricetin as predominant flavonoids, and proline, methionine, and arginine as dominant amino acids. Even though Mahuad plants (LRL) are widely grown in Thailand and other Southeast Asian countries, they are still underutilized and lack widespread commercial cultivation. Therefore, further comprehensive studies are needed to establish strong scientific evidence as well as assess the toxicity levels of LRL seeds, guiding future directions for seed utilization. Our research takes a pioneering step in quantifying these individual bioactive compounds and creating comprehensive profiles of LRL seeds, providing a foundational resource for further exploration across various fields and applications.

**Supplementary Materials:** The following supporting information can be downloaded at: https://www.mdpi.com/article/10.3390/horticulturae9101159/s1, Figure S1. The chromatogram of amino acids in both standard and LRL seed samples from LC/MS/MS; Figure S2. The HPLC chromatogram of sugars and oligosaccharides including stachyose, fructooligosaccharide, sucrose, and glucose in LRL seed.

**Author Contributions:** Conceptualization, S.S. (Sirithon Siriamornpun); methodology, T.C., S.S. (Surapon Saensouk), A.B. and S.S. (Sirithon Siriamornpun); validation, A.B. and S.S. (Sirithon Siriamornpun); investigation, S.S. (Sirithon Siriamornpun), T.C. and A.B.; resources, T.C. and S.S. (Surapon Saensouk); writing—original draft preparation, A.B., S.S. (Sirithon Siriamornpun) and T.C.;

writing—review and editing, A.B., S.S. (Surapon Saensouk), T.C. and S.S. (Sirithon Siriamornpun); supervision, S.S. (Sirithon Siriamornpun); project administration, S.S. (Sirithon Siriamornpun); funding acquisition, S.S. (Sirithon Siriamornpun). All authors have read and agreed to the published version of the manuscript.

**Funding:** This research project was financially supported by Thailand Science Research and Innovation (TSRI).

**Data Availability Statement:** Data are contained within the article.

**Acknowledgments:** This research project was financially supported by Thailand Science Research and Innovation (TSRI) and Mahasarakham University. The authors thank the Laboratory Equipment Center of Mahasarakham University for cooperation and scientific assistance. Also, thanks to Colin Wrigley, University of Queensland, Australia, for English proofreading.

**Conflicts of Interest:** The authors declare no conflict of interest.

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
