# Peer review of "Bioactive Compounds, Amino Acids, Fatty Acids, and Prebiotics in the Seed of Mahuad (Lepisanthes rubiginosa (Roxb.) Leenh)"

_horticulturae, doi:10.3390/horticulturae9101159_

Round 1
Reviewer 1 Report
The manuscript is well-prepared and organized. Conclusions are supported by the results. However, in my opinion, the manuscript suffers serious flaws in experiments that need answers:
1. Why did you use HPLC for analysis of polyphenols? They are found in very low concentrations, and HPLC-MS/MS should be used here.
2. How did you identify compounds in both analyses (polyphenols and amino acids)? How the quantification was performed?
3. Why did the authors not isolate volatile compounds (terpenes) and their analysis? In my opinion, the presence or absence of these compounds is also an important result.
3. Where is the analysis of the fatty acids? They are the primary source of energy in seeds. It is of the highest importance to know the content of fatty acids.
based on the above comments, I suggest a major revision of the manuscript with the additional experiments before the reconsideration of this manuscript for publication.
The English language is very good. The soundness of the manuscript is also very good. I suggest a minor revision of small mistakes in grammar and syntax.
Reviewer 2 Report
The paper presents proximate composition, phenolic acids, flavonoids, amino acids, sugars, and oligosaccharides contents, and FTIR spectrum of Lepisanthes rubiginosa (Roxb.) Leenh seed. The information is new and relevant, as this seed is little exploited, and could become a food and feed source.
Overall, the paper is well written and well discussed, except for the FTIR results.
English needs editing. Although the paper is mostly well written, there are some grammar flaws.
The major problem is on statistics. It makes no sense to apply ANOVA to different components of the same sample (phenolic acids, flavonoids, amino acids, and sugars). The compounds have no reason to be compared among them, as there were no different treatments applied so that their influence on the result could be evaluated. Therefore, I suggest presenting results as simply mean ± SD.
I recommend accepting the paper after minor revision. More specific comments follow next.
Abstract
Line 25: “future use in various aspects and applications” must be specified. What are the possible applications of LRL seeds?
Material and methods
Lines 70-71: only four flavonoids were named
Lines 98-99: Which AOAC methods were used? It needs specification, as there are several methods to determine each of the components.
Line 124: 1 mL of solvent was enough to obtain amino acids from 100 g of sample? This number seems odd.
Line 125: v/v ratio? Specify.
Results
Table 1: The SD should be reported as only one figure, which must coincide with the last significant figure of the value. It applied to all tables. If the doubt is in determined figure, it makes no sense to consider the subsequent figures, as they are all doubtful and less significant than the first one. The SD does not have to be reported with the same number of decimal places for all values in the table; it has to be reported in the decimal place it if meaningful.
Table 2: To me it makes no sense to apply ANOVA for different components of the same sample. It would have the same effect of applying it no proximate composition of Table 1. The compounds have no reason to be compared among them, as there were no different treatments applied so that their influence on the result could be evaluated.
Table 3: idem.
Table 4: idem.
Discussion
Lines 339-342: What insights on the chemical composition and structure of LRL seeds were you able to extract from the FTIR results?
English needs editing. Although the paper is mostly well written, there are some grammar flaws.
Reviewer 3 Report
This manuscript studied the bioactive compounds, amino acids and prebiotics in the seed of Mahuad. According to the data obtained in the present study, vanillic acid, p-hydroxybenzoic acid ,quercetin, fructooligosaccharide were analyzed in the seed of Mahuad .This work is interesting and valuable. I think this manuscript is acceptable after addressing the following questions. The comments are listed as follows:
1. The introduction is too short and not well focused. The section needs to be elaborated in particular the authors should highlight the novelty of this work, and particularly illustrate the superiority of this work from previous reports about bioactive compounds, amino acids and prebiotics, especially the seed of Mahuad.
2. Line 40-indentation for the first line.
3. Line 123 - 133-Which column was used? Which eluent gradient?
4. What's the purpose of section 3.5? Is there any different results compared with the specific synthesis of vanillic acid, p-hydroxybenzoic acid, quercetin;,fructooligosaccharide ? If so, why?
5. Line 339 - 342- The Discussion section lacks scientific depth. How is this work different from the paper published in other journals at recent years? The sections should be critically discussed and compared with the previous reports.
6. The conclusion section needs improvement. What are the authors' own viewpoints? What are the major findings and how they are addressing the left behind research gaps and current challenges? In conclusion, it is difficult to recognize the novelty in this article that is sufficient for publication.
